# Management of Digital Dermatitis in Dairy Herds: Optimization and Time Allocation

**DOI:** 10.3390/ani13121988

**Published:** 2023-06-14

**Authors:** Rodolphe Robcis, Ahmed Ferchiou, Mehdi Berrada, Didier Raboisson

**Affiliations:** 1Unité Mixte de Recherche, Animal Santé Territoires Risques Ecosystèmes, Centre de Coopération Internationale en Recherche Agronomique pour le Développement, 34000 Montpellier, France; ahmed.ferchiou@envt.fr (A.F.); mehdi.berrada@envt.fr (M.B.); didier.raboisson@envt.fr (D.R.); 2Animal Santé Territoires Risques Ecosystèmes, Centre de Coopération Internationale en Recherche Agronomique pour le Développement, Institut National de Recherche Agronomique et Environnement, University of Montpellier, 34090 Montpellier, France; 3Ecole Nationale Vétérinaire de Toulouse, Université de Toulouse, 31300 Toulouse, France

**Keywords:** lameness, cow, economics, herd management, footbath, animal welfare

## Abstract

**Simple Summary:**

Foot diseases are a widespread issue in dairy herds that lead to economic losses and cow welfare alteration. Time is a limited key resource for farmers in herd management and especially for the control of lameness. The aim of this study is to determine the most effective time to be allocated for digital dermatitis management and to quantify the marginal economic gain for extra time devoted to digital dermatitis management in dairy herds. This study shows that allocating less time to lameness detection and more to footbath application is more profitable for both farmers and animal welfare. The optimal time to spend on footbath application ranges from 17.8 to 22.3 h/month.

**Abstract:**

The objective of this study is to determine the most effective time allocation for digital dermatitis management and to quantify the marginal economic gain from extra time devoted to digital dermatitis management. The model simulating foot disease occurrence and the associated management was Dairy Health Simulator©. Then, an econometric model was applied to identify the relationship between the dairy workshop’s gross margin and time for lameness management as well as the hourly marginal gain curve associated with lameness management. The gross margin was optimized under two constraints, i.e., the overall time spent for lameness management and the mean lameness duration (mimicking cow welfare). The results show that allocating less time to lameness detection and more to footbath application can contribute as follows: (i) reduces time spent for lameness management, (ii) maintains the best welfare level, and (iii) obtains the highest gross margin. The optimal time to devote to footbath application ranged from 17.8 to 22.3 h/month. A hiring strategy was investigated, and the break-even point ranged from 16.1 to 19.8 h/month. The recommended time to spend on footbath application is relatively important; therefore, stakeholders should consider the importance of lameness to the dairy industry and should devote enough time for footbath applications.

## 1. Introduction

Foot diseases are one of the most important concerns in dairy herds [1,2,3]; studies in the literature describes a wide range of prevalence, from 5% to 55% [4,5,6,7,8]. Foot diseases lead to deterioration in cow welfare, reduced feed intake, severe pain, arched back, as well as abnormal posture and walking [9,10]. Foot diseases also cause major economic losses, through treatments, reduced milk production, discarded milk due to antimicrobial use, deteriorated reproductive performance, early culling, and interactions with other diseases such as subclinical ketosis and mastitis [11,12,13]. The estimations, available in the literature, of the costs associated with lameness, including all types of etiologies, range between 43 and 1040 euros/case/year [2,5,11,13,14,15,16]. Although from 27% to 38% of farmers only consider foot diseases as moderate or large problems in herds [17,18], lameness management is one of the most important challenges for farmers [17,19]. Management of lameness relies on different procedures at the cow level, such as trimming or use of non-steroidal anti-inflammatory and antimicrobial drugs [20], which require observational time to detect lame cows. Short-term herd-level actions include increasing scraping frequency to improve hygiene and to control infectious foot diseases such as interdigital dermatitis and digital dermatitis (DD), and applying a footbath (FB) to treat DD [21,22,23]. Mid-term actions notably include systematic preventive trimming for cows at dry-off concomitantly to curative trimming chutes for lame cows. Long-term actions are about housing design, especially the comfort of cubicles and the nature of the floor that influences pressures applied on the claws [24,25]. All these measures are costly and time-consuming for farmers [26]. In this context, decision making should account for multiple factors, time being very often a limiting factor. Time devoted to lameness management can be divided into two main categories, i.e., time spent for detection of lame cows and time spent for treatment. Lameness detection is most often performed by visual inspection of cows. The five-scale lameness score classification is a useful tool for evaluating the degree of lameness in cows [27]. This classification is based on observations of the posture and position of the limbs, and it requires both observation time as well as experience and some technical knowledge to rightly classify the cows. Lameness detection consequently depends on the degree of lameness [28], farmers’ skills, and time allocated to this task. The combination of these three aspects generates different levels of expertise in the detection of lame cows, which has already been investigated in a previous study [29]. On dairy farms, lameness is often underestimated [18,30,31] because of a lack of consistent knowledge on the subject, especially about the different degrees of lameness and the ability to detect all of them, particularly the most discrete cases [4,27,30,31]. Underestimation ranges from 17% to 30% [32,33] and induces negative outcomes for farmers. A previous study showed that one extra week spent in a lame state for one cow cost 12.1 euros [29].

Lame cows can also be automatically detected thanks to image-processing techniques, kinetic or kinematic processes, often with poor specificity for early detections [34]. Infrared thermography is a technique for detecting lame cows by identifying foot-located inflammatory processes through the assessment of surface thermal responses under various conditions [35]. However, this new and non-invasive technique also lacks specificity in identifying lame cows [36]. Time spent for treatment includes time spent for trimming chutes, antibiotic administration, and collective FB application to specifically treat active lesions of DD, which is recommended when the prevalence of DD exceeds 20% [37]. Between 0 and 96.7% of dairy herds are challenged by DD occurrence worldwide [38], and the majority of time spent for treatment is widely devoted to FB application [18]. Footbath application is, therefore, one critical point in the time spent for treatment of lame cows. As a result, early detection of lame cows as well as subsequent, prompt, and appropriate treatment are critical points to minimize losses and to improve animal welfare. Inspections and the time for treatment are both considered to be time-consuming by farmers [39].

A better understanding of substitutions for these actions devoted to lameness management in order to increase financial performances of farms while improving animal welfare and limiting extra labor is of interest to the dairy industry. The objective of the present study is to determine the most effective time to be allocated for DD management and to quantify the marginal gain for extra time devoted to DD management in dairy herds.

## 2. Materials and Methods

### 2.1. Dairy Health Simulator

DHS© (Dairy Health Simulator©) is a bioeconomic model aimed at supporting decision making for farmers in French dairy herds [40]. The model consists of a biological simulation model coupled with an economic optimization model. The biological model is defined on a cow-week basis and on the weekly probabilities for all cow events, including milk production, reproduction, and diseases. It aims to achieve a long-term dynamic representation of a dairy herd of 100 cows. In brief, from birth to death, each animal is characterized weekly by their physiological and production status (e.g., male calf, female calf, pregnant, in-milk cow, and dry cow).

A specific module (DHS_Lame), which had been previously described in the literature, was built to accurately simulate the occurrence and consequences of 5 claw disorders, i.e., sole ulcer, white line disease, interdigital phlegmon, interdigital dermatitis, and DD [29]. M-stage dynamics were also implemented [41,42] in order to accurately simulate DD occurrence (Figure 1). Existing interactions between foot diseases were included [42], as well as interactions with other diseases such as mastitis and subclinical ketosis [43]. Each claw etiology is specifically responsible for subsequent milk yield losses [34], and each claw disorder generates a certain degree of lameness quantified by a 5-scale lameness score classification [27]. Each level of lameness score is responsible for reproductive disorders such as increased luteal phase duration [44] and a lack of heat expression [29]. The farmer-level ability to detect lame cows depends on the lameness score [28] and was implemented in the model. Once cows were detected as lame, appropriate treatments (claw trimming, including foot block application for white line disease, sole ulcer and interdigital dermatitis, systemic antibiotics for interdigital phlegmon, oxytetracycline spray, or collective FB for DD, depending on the herd-level DD prevalence [37]) to cure them were applied leading to healing with a given probability. The appropriate treatments were applied either by a veterinarian, a trimmer, or by the farmer himself. The choice of the person applying the treatment was governed by in-field-based probabilities in this model. As long as lame cows were not detected and subsequently treated, they remained lame and negative effects were still applied.

A simulated stabilized 100 in-milk cow herd was used as a unique starting base for all scenarios for a 728-week simulation with 100 iterations each. The last 520 weeks were included in the results analysis to obtain stable results. The results of the preliminary study showed a mean lameness prevalence of 55%. Relative contributions of DD, interdigital dermatitis, interdigital phlegmon, sole ulcer, and white line disease were 36%, 28%, 4%, 19%, and 13%, respectively. The mean herd-level prevalence of DD was 28% [29].

### 2.2. Outcomes and Metrics

The medium lameness duration (MLD) [29] represents the mean period during which one given cow remains lame, i.e., the number of weeks during which one cow exhibits a lameness score greater than or equal to 3. This parameter is expressed in weeks and ranges from 4.6 to 20 weeks, depending on the different in-farm situations [29].

The milk workshop’s gross margin (GM) was calculated as the difference between gross sales and workshop’s operational costs (Equation (1)):(1)$${GM}_{s}={\left({MilkR}_{s}+{MeatR}_{s}+{AnimR}_{s}\right)}_{}-{\left({FeedE}_{s}+{ReproE}_{s}+{VetE}_{s}+{TrimE}_{s}+{TreatE}_{s}+{StrawE}_{s}\right)}_{}$$
where MilkR*_S_*, MeatR*_S_*, and AnimR*_S_* are revenues from milk, meat production, and sold animals (including claves and pregnant heifers), respectively, and FeedE*_S_*, ReproE*_S_*, VetE*_S_*, TrimE*_S_*, TreatE*_S_*, and StrawE*_S_* are expenses from feeding cost, reproduction cost, veterinary cost, treatment cost, trimmer cost, and strawing cost, respectively.

AllLameTime included time spent in detecting lame cows for each different lameness detection rate LD_*i* (1 ≤ *i* ≤ 11) (DetectTime), time dedicated for curative trimming (TrimTime), time devoted to antibiotic injection (ATBTime), time dedicated to topical treatment (TopicTime), and time devoted to FB application for each DD prevalence threshold FB_*j* (0 ≤ *j* ≤ 9) from which the FB is applied (FBTime) (Equation (2)):(2)AllLameTime=DetectTime+TrimTime+ATBTime+TopicTime+FBTime

Time dedicated to lameness management (AllLameTime) was classified into five classes, mimicking five different scenarios corresponding to farmers’ commitments. Each month, farmers could spend less than 7 h (poor implication), between 7 and 14 h (low implication), between 14 and 21 h (moderate implication), between 21 and 28 h (good implication), and between 28 and 35 h (excellent implication) for lameness management. Based on the range highlighted in a previous study and discussed above [29], five classes of MLD were generated, representing five different levels of lameness-related cow welfare as follows: between 4 and 8 weeks (excellent level of cow welfare), between 8 and 11 weeks (good level of cow welfare), between 11 and 14 weeks (moderate level of cow welfare), between 14 and 17 weeks (low level of cow welfare), and between 17 and 20 weeks (poor level of cow welfare).

### 2.3. Management Scenarios

The dynamics among the foot diseases were simulated stochastically and holistically leading to a panel of lameness prevalence as a result of different combinations of scenarios: flooring (concrete (CONCRETE) vs. textured (TEXTURED)), scraping frequency (≤8 times per day (SCRAP ≤ 8) vs. >8 times per day (SCRAP > 8)), and existence of preventive trimming (yes (TRIM) vs. no (NOTRIM)) (Table 1). Moreover, a standard level of lameness detection corresponding to 30 min spent per day was simulated (LD_6) by referring to the literature [28]. Ten other levels of detections (5 below and 5 above) were simulated, from no detection of lame cows at all (LD_1) to detection two times better than a standard farmer (LD_11) (Table 2). The prevalence of DD that led to footbath application was 20% [37] in the base scenario (FB_4) and simulations included steps of 5% of the threshold, from footbath systematically used to whatever the DD prevalence (FB_0) to footbath never applied (FB_9) (Table 3).

These different sets of scenarios and their combinations brought a wide range of prevalence for lameness and DD.

### 2.4. Optimization under Constraints and Econometrics

The optimization under the two constraints is schematically represented in Figure 2. The intersection between the two groups ([a ≤ AllLameTime ≤ b] and [c ≤ MLD ≤ d]) gave the group in which all the GM favorably responded to the two constraints ([a ≤ Time ≤ b] ∩ [c ≤ MLD ≤ d]), where a, b, c, and d are applied thresholds, see above. Then, among these GM, the two maximum values (*max_1_*(GM) and *max_2_*(GM)) were elected according to Formulas (3) and (4):(3)$${max}_{1}(GM),\forall GM\in [a\leq Time\leq b]\cap [c\leq MLD\leq d],GM\leq max1(GM)$$
(4)max2(GM),∀GM∈[a≤Time≤b]∩[c≤MLD≤d]\{max1(GM)},GM≤max2(GM)≤max1(GM)}

From the 25 subsequent levels of constraints (5 AllLameTime * 5 MLD), the two best GMs for each of them were selected. Two levels were considered to keep variability within the results of the combination of scenarios. An econometric model was applied to quantify the association between the time spent for FB application and the GM and to calculate the corresponding marginal economic gain of time spent. The raw distribution of data describes a sum of quadratic and linear terms [52] and the general form of our production function is:(5)$$Y=a+bX-cX^{2}$$

The simple quadratic equation with a minus sign before c denotes a decrease in marginal returns. The optimum can be obtained by deriving the production function and resolving Equation (6):(6)$$0=b-c2X_{*}$$
where $X_{*}$ is the solution, thus the optimum value. Substituting the optimal value of *X* in Equation (5) leads to Equation (7):(7)$$Y=a+bX_{*}-cX_{*}^{2}$$

The inclusion of a hiring strategy was also explored. Assuming the quadratic function governing the evolution of the marginal gain as a function of the time spent on FB application (see above), the break-even point of employment for a farm worker was calculated by comparing it to the marginal hourly gain obtained in each scenario and based on a gross hourly income of 11.27 euros (French Agricultural Chamber basis).

The break-even point *X_BEP_*, obtained for an hourly marginal gain equal to the employee’s gross hourly income, is then calculated according to Equation (8):(8)$$b-2cX_{BE_{P}}=11.27$$

## 3. Results

For the 25 different combinations of constraints (AllLameTime and MLD), the best and second-best results of scenarios are presented in Table 4. For instance, for the lowest classes of MLD and AllLameTime (upper left part of the table), the two optimal scenarios are “CONCRETE_TRIM_SCRAP>8_LD_2_FB_0” and “CONCRETE_TRIM_SCRAP>8_LD_2_FB_1”, corresponding to the scenarios in which the floor is concrete, preventive trimming is performed, daily scraping frequency is above eight times, lameness detection is 80% lower than the standard implemented one [28], and the DD prevalence for which a collective FB is applied is 0% or 5%. The color code helps to classify the scenarios from the less acceptable (red) to the most acceptable (green), and helps to make the interpretation easier.

More desirable scenario. Lower levels of welfare (i.e., MLD > 11 weeks, half-right side of Table 4) are observed in the context where scenarios of flooring, trimming, and hygiene are variable (red/green), whereas MLD ≤ 11 weeks is almost always associated with the most acceptable scenarios CONCRETE, TRIM, and SCRAP>8 (green). Most of the change in MLD is linked to lameness detection and FB application scenarios. Scenarios associated with “good practices” in FB application (i.e., low numbers of classes) are observed in classes of low MLD, and vice versa. Scenarios associated with “good practices” in lameness detection (i.e., high numbers of classes) are also observed in classes of low MLD, and vice versa.

The results show that the outcome AllLameTime is highly sensitive to the constraint DetectTime (change of color with a row for lameness detection), whatever the level of MLD. On the contrary, the threshold for FB application is not identified as a key parameter influencing AllLameTime, for the best class of MLD (between 4 and 8 weeks) because classes for time spent for FB application are always 1 or 2. On the contrary, AllLameTime is sensitive to the threshold for FB application (change of class in a given column), especially for the MLD class between 8 and 11 weeks, where FB application classes range from 2 to 9. To sum up, an interaction-like behavior is described for the variable threshold for FB application with the variables MLD and AllLameTime.

The Tornado diagram describing GM (highest at top) for 24 combinations of FB and LD (Figure 3), in the context where the panel of obtained GM is the greatest (i.e., where risk factors are effectively managed, including concrete floor, high daily scraping frequency, and existence of preventive trimming), shows that the seven best GMs are associated with low levels of lameness detection (i.e., LD_1 and LD_2,) and that the GM increases with an increase in FB (from FB_7 to FB_0).

The econometric analysis of the relationship between the time spent for FB application in the context of DD management and GM for the eight combinations of scenarios (CONCRETE_TRIM_SCRAP>8, CONCRETE_TRIM_SCRAP≤8, CONCRETE_NOTRIM_SCRAP>8, CONCRETE_NOTRIM_SCRAP≤8, TEXTURED_TRIM_SCRAP>8, TEXTURED_TRIM_SCRAP≤8, TEXTURED_NOTRIM_SCRAP>8 and TEXTURED_NOTRIM_SCRAP≤8) (Figure 4) strongly follows a second-degree polynomial (r-squared ranging from 0.74 to 0.84). Depending on the combination, the optimized GM is reached for overall time spent for FB application ranging from 17.8 to 22.3 h/month (Table 5). The highest values of GM correspond to a zero marginal gain, as illustrated in Figure 5 and Table 5 (see below). Whatever the combination of scenarios observed, the optimum is observed on the right of the graph, showing the high economic interest of spending time on FB application to manage DD.

The marginal gain of time devoted to FB application (Figure 5) represents the value in euro of one extra hour spent on FB application. It is obtained by calculating the derivative function of the corresponding quadratic function for each combination of scenarios. The marginal gain is higher in the case of SCRAP≤8 (daily scraping frequency ≤8 times) compared to SCRAP>8 (daily scraping frequency >8 times) (from 5% to 21.9%), higher in the case of TRIM (preventive trimming) compared to NOTRIM (no preventive trimming) (from 7.4% to 48.4%), and higher in the case of TEXTURED (textured floor) compared to CONCRETE (concrete floor) (from 18.6% to 55.8%).

The optimal time of use of the FB as well as the marginal gain in time spent for FB application was determined based on a farmer exclusively working on his own. The possibility of hiring an employee was also studied to determine the possible profitability of such an investment.

First, the optimal time to spend on FB application ranged from 17.8 to 22.3 h per month, which represented between 12.8% and 15.9% of the total monthly working time of a full-time employee (based on a working week duration of 35 h). In other words, the optimal time devoted to FB application could be achieved by a half-time worker working approximately one week per month.

The break-even point was calculated for all the combinations of scenarios. The results are presented in Table 6. Depending on the combinations of scenarios, the break-even point ranged from 16.1 and 19.8 h per month (corresponding to the combinations of scenarios TEXTURED_NOTRIM_SCRAP>8 and CONCRETE_TRIM_SCRAP≤8, respectively). In other words, this break-even point is the maximum amount of time that an employee can be hired per month in order to keep the profitability of spending time of FB application.

## 4. Discussion

Regardless of the farm context, spending time to detect lame cows and to treat them is crucial, and there is a need to consider both measures in dairy herds. Digital dermatitis plays a major role in total lameness cases, with a high correlation among herd-level DD prevalence, lameness prevalence, and MLD [29]. The incidence rate of lesion development has been reported to be four lesions per 100 cow foot-months on average [53]. *Treponema* spp. have been implicated as the causative agents in DD due to their identification in DD lesions [54]. However, the in-field diagnosis has mainly been based on the visual inspection of the feet during trimming sessions [55]. This inspection method has been adopted as a gold standard method by many researchers [56,57]. Management of DD requires implementation of collective FBs at the herd level [22,23]. However, lameness detection and FB application are time-consuming measures, in the context where time is already a scarce resource for the farmer. As a result, the time spent should be effectively allocated to obtain the best profitability as possible.

The present results confirm that the overall time spent for lameness management (AllLameTime) is highly sensitive to the constraint time spent for lameness detection (DetectTime). Lameness detection has already been identified as a time-consuming task [18,39,58] and lameness detection through lameness 5-scale scoring has low sensitivity, leading to low economic value attached by farmers to their application [59]. Acute cases of lameness induce major inflammatory processes explaining severe pain, such as increased levels of haptoglobin and proinflammatory cytokines [60]. If lameness is not detected early, acute inflammatory processes turn into chronic stages and have systemic repercussions with hormonal dysfunction leading to reproductive disorders [60]. Moreover, the effectiveness of any type of treatment is significantly reduced in the case of chronic cases [61]. As a result, acute cases of lameness turn into chronic stages and lead to negative consequences on reproductive performance (e.g., an increase in metritis prevalence [62] and an increase in luteal phase duration [44]). As with the management of chronic mastitis, chronic cases of lameness should be included in culling strategies to improve herd-level animal welfare and to reduce economic losses to the farmer [61].

Spending less time on lameness detection but more time on footbath application can result in the following: (i) reducing overall time for lameness management (AllLameTime), (ii) maintaining good welfare level (MLD), and (iii) obtaining the highest margin in the set of combinations of scenarios (Figure 5). The easiness of FB application, without negative side effects, combined with economic benefits demonstrates that FB application is a highly profitable practice. The therapeutic and preventive efficacies of FB in DD cases have been widely discussed in the literature. The choice of the active principle in the FB preparation is also a point of discussion. A meta-analysis showed that a 5% CuSO_4_-based FB used at least four times a week was the only significantly effective protocol for the curative treatment of DD [22]. Preventive efficacy was not proven, and therefore, it was not included in the calibration. Applying a FB is an easy routine with a systematic protocol to follow, and should be promoted for both the farmers’ and the cows’ welfare. Spending time on detection of lame cows requires more skills to determine cows’ lame status and behaviors. It also requires a time slot specifically dedicated to this task, which is not always respected in practice and additionally with poorer results [59].

However, in practice, the separation of tasks is not so obvious, and the allocation of time is not performed in a strict mechanical way. In general, lameness detection is performed through the observation of cows for parameters related to reproduction (e.g., heat detection) or any other animal health issues. Here, we have a spillover effect, known as the health impacts and costs that extend beyond a health intervention or program’s targeted recipient to unintentionally impact other recipients either in a positive or negative way [63]. This aspect inevitably induces a bias in the allocation of a farmer’s time and it is a limitation of the present study.

The econometric approach shows that the optimized GM is reached for overall time spent on FB application ranging from 17.8 to 22.3 h/month for the present 100-cow simulated dairy herd, depending on the scenarios (Figure 5 and Table 5). To the best of our knowledge, this is the first study to investigate the relationship between time allocation for FB application and producers’ profits. Previous studies have proposed standardized protocols based on prevalence thresholds to manage digital dermatitis [23,37].

We also quantify the marginal gain of time spent for FB application (Figure 5). We chose to calculate marginal gain for its ability to display where to prioritize efforts. First, we show that FB application and preventive trimming act in synergy, and that combining these two practices is economically justified. Indeed, on the one hand, trimming contributes to good preventive management of foot diseases [64], especially the non-infectious ones such as sole ulcer and white line disease [48] and concomitantly lead to the identification of DD lesions and to subsequent treatment at the cow level. On the other hand, footbath application is the conventional collective treatment for DD, which is the most important infectious foot diseases in dairy herds [38]. As a result, the synergy is two-fold as it allows for a combination of individual and collective treatments, and it also includes both infectious and non-infectious diseases in the general lameness management.

Second, we show that the marginal gain of time spent on FB application is more important when the floor is textured, although textured floors increased the relative risk for claw disease. This can be interpreted as a substitution effect altogether acting as a compensation. Third, the marginal gain of time spent on FB application is more important in the case of limited hygiene (SCRAP≤8). A lack of hygiene is known to be a critical point in foot disease occurrence, especially in the case of digital dermatitis [47,65]. This suggests that FB application may have a substitution effect on those deteriorated scenarios. Even though FB has a better marginal gain in the case of limited hygiene, our results do not suggest that limited hygiene should be promoted, since the highest GM is systematically observed in the scenario with SCRAP>8 (Table 4).

Whatever the combination of scenarios, hiring an employee is an effective option for the farmer, both to keep benefits in time devoted to FB application and to gain time which should be allocated to other crucial tasks such as heat detection, feeding plans, or housing management. The calculated break-even points will be helpful for farmers in order to achieve the best herd management as possible, integrating both profitability and time allocation.

The key variables used in the present study are evidenced based, and the final choices of variables to be included in the study were based on the literature and author’s experience. Previous studies have highlighted that preventive trimming is beneficial and have shown that routine trimming reduced lameness cases by 25–34%, especially non-infectious foot diseases [45,50]. However, because of a lack of confidence regarding the benefits, only 12.9% of farms have adopted preventive trimming [18]. It is critical to bring economic arguments to veterinarians to convince farmers to adopt best management practices.

There are some limitations to the present study. Other etiologies of lameness, such as interdigital hyperplasia or painful lesions regarding proximal limb regions, may be included in the model. The chronic lameness cases discussed above should also be included in the model, both in the treatment decision strategies and in the culling strategies. Future investigations should, therefore, be carried out in order to have a model that is even more representative of field conditions.

## 5. Conclusions

In the context of digital dermatitis management, this study demonstrates the economic benefit of allocating time to footbath application to maintain both good welfare level and gross margin, and time devoted to observation can be reduced in such a perspective. The optimized gross margin is reached for the overall time spent on footbath application ranging from 17.8 to 22.3 h/month, allowing a certain hiring policy, with an upper limit ranging from 16.1 to 19.8 h per month, and optimizing both benefits and farmer-related time allocation. The combination of footbath application and preventive trimming is economically relevant.

## Figures and Tables

**Figure 1 animals-13-01988-f001:**
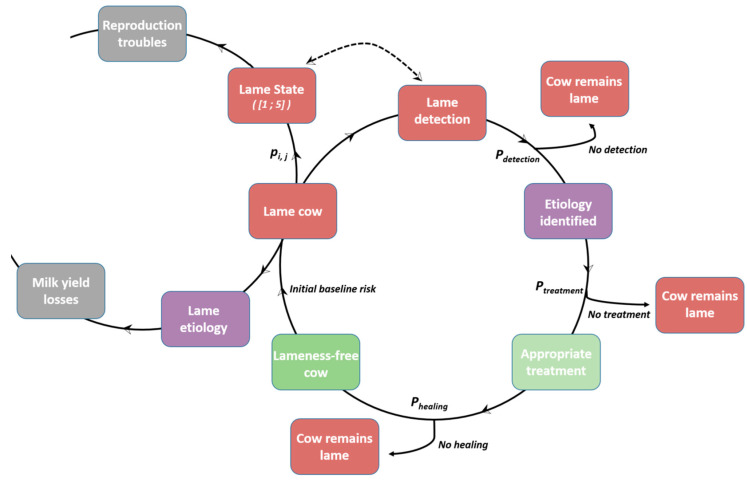
Schematic summary of lame disease calibration [29]. A cow becomes lame with an initial baseline risk for each etiology, and the degree of lameness (5-point scale) is determined with a probability p_i, j_. Once the cow becomes lame, the etiology contributes to the subsequent milk yield losses, whereas the lame state generates reproductive disorders. The detection by a farmer is based on his/her ability to detect the lame state (arrows with dotted line) (P_detection_). Once the cow is detected as lame, the etiology is systematically correctly identified, and a probability of treatment (P_treatment_) is applied. Then, the probability of healing (P_healing_) is applied to determine whether the cow remains lame.

**Figure 2 animals-13-01988-f002:**
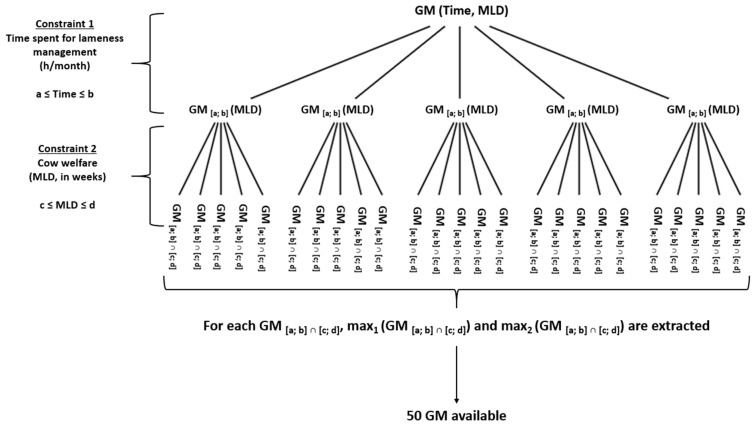
Optimization under two constraints. GM, gross margin and MLD, medium lameness duration. a, b, c, and d represent thresholds for the applied constraints. [a,b] ∈ {[4, 8], [8, 11], [11, 14], [14, 17], [17, 20]}. [c,d] ∈ {[0, 7], [7, 14], [14, 21], [21, 28], [28, 35]}.

**Figure 3 animals-13-01988-f003:**
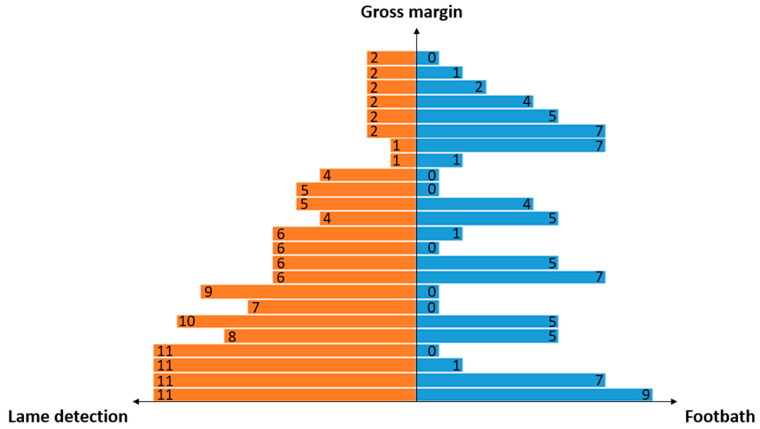
Tornado diagram illustrating the sensitivity analysis about the influence of time devoted to lameness detection and time spent for footbath application, in the context where risk factors are well-managed (concrete floor, existence of preventive trimming, and high daily scraping frequency (CONCRETE_TRIM_SCRAP>8)). Numbers correspond to the different levels of scenarios. LD: from 1 (no time devoted to lameness detection) to 11 (60 min devoted to lameness detection) with a step of 5 min between each level. Details in Table 2. FB: from 0 (FB systematically applied whatever the digital dermatitis prevalence) to 9 (FB never applied) with a prevalence step of 5%. Details in Table 3.

**Figure 4 animals-13-01988-f004:**
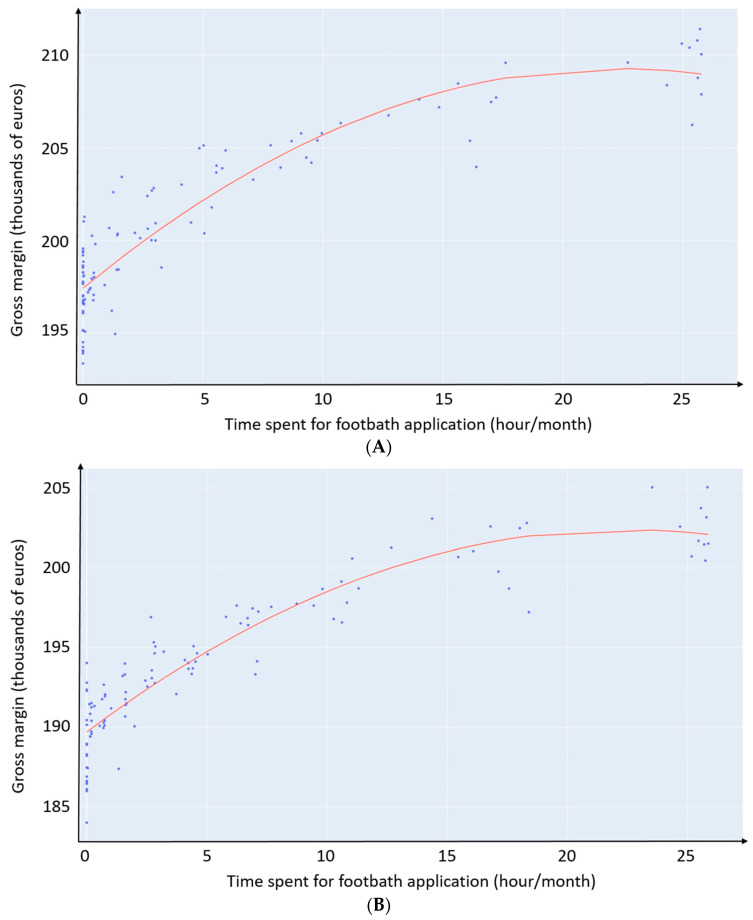
Relationship between time spent for footbath (FB) application and gross margin (GM): (**A**) CONCRETE_TRIM_SCRAP>8; (**B**) CONCRETE_TRIM_SCRAP≤8; (**C**) CONCRETE_NOTRIM_SCRAP>8; (**D**) CONCRETE_NOTRIM_SCRAP≤8, (**E**) TEXTURED_TRIM_SCRAP>8; (**F**) TEXTURED_TRIM_SCRAP≤8; (**G**) TEXTURED_NOTRIM_SCRAP>8; (**H**) TEXTURED_NOTRIM_SCRAP≤8. CONCRETE, concrete floor; TEXTURED, textured floor; TRIM, preventive trimming; NOTRIM, no preventive trimming; SCRAP>8, daily scraping frequency >8 times; SCRAP≤8, daily scraping frequency ≤8 times.

**Figure 5 animals-13-01988-f005:**
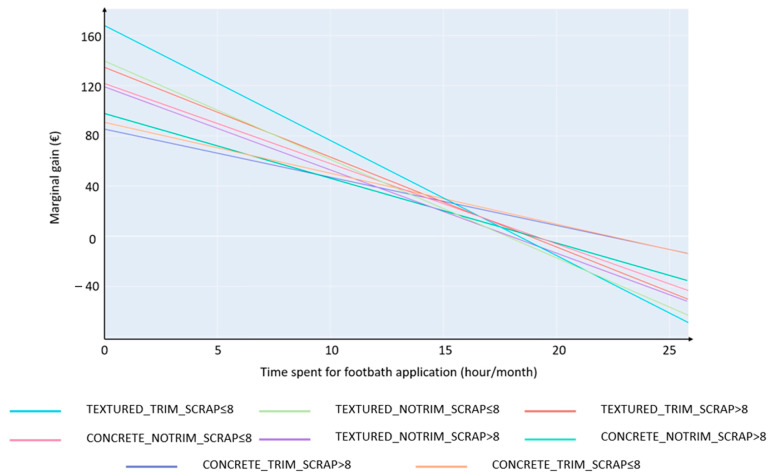
Marginal gain relative to time spent on footbath application, for each combination of scenarios. TEXTURED, textured floor; CONCRETE, concrete floor; TRIM, preventive trimming; NOTRIM, no preventive trimming; SCRAP≤8, daily scraping frequency ≤8 times; SCRAP>8, daily scraping frequency >8 times.

**Table 1 animals-13-01988-t001:** Scenarios influencing the initial baseline risk of claw disease occurrence (adapted from [29]).

		Risk Factors			
		Sole Ulcer	White Line Disease	Digital Dermatitis	Interdigital Dermatitis
FLOORING	Concrete (“CONCRETE”)	reference	reference	reference	reference
	Textured (“TEXTURED”)	2 [45]	2.5 [46]	2.7 [47]	2.7 [47]
HYGIENE Scraping frequency	>8 times a day (“SCRAP>8”)	reference	reference	reference	reference
	≤8 times a day (“SCRAP≤8”)	1	1	2 [48]	2 [48]
PREVENTIVE TRIMMING	Yes (“TRIM”)	0.7 [49]	0.7 [50]	1	0.8 [51]
	No (“NOTRIM”)	reference	reference	reference	reference

**Table 2 animals-13-01988-t002:** Scenarios on lame detection (LD) by farmers.

Detection Rate Scenarios	Multiplier Coefficients	Corresponding Time Spent in Detecting Lame Cows (min/Day)
LD_1	0	0
LD_2	0.2	5
LD_3	0.4	10
LD_4	0.6	15
LD_5	0.8	20
LD_6 ^(1)^	1	30
LD_7	1.2	35
LD_8	1.4	40
LD_9	1.6	45
LD_10	1.8	50
LD_11	2	60

^(1)^ inspired by [28].

**Table 3 animals-13-01988-t003:** Scenarios relative to digital dermatitis (DD) prevalence threshold from which a collective footbath (FB) is applied.

Footbath Use Scenarios	DD Prevalence Thresholds (%)
FB_0	0
FB_1	5
FB_2	10
FB_3	15
FB_4 ^(1)^	20
FB_5	25
FB_6	30
FB_7	35
FB_8	4
FB_9	110

^(1)^ Standard reference of footbath application [37].

**Table 4 animals-13-01988-t004:** Optimization of the gross margin under constraints of time spent for lameness management and lameness-related cow welfare. Numbers correspond to the different levels of scenarios for lameness detection (LD) and footbath (FB).

		Medium Lameness Duration (Weeks)																								
		[4; 8]					[8; 11]					[11; 14]					[14; 17]					[17; 20]				
		FLOOR	TRIMMING	HYGIENE	LD	FB	FLOOR	TRIMMING	HYGIENE	LD	FB	FLOOR	TRIMMING	HYGIENE	LD	FB	FLOOR	TRIMMING	HYGIENE	LD	FB	FLOOR	TRIMMING	HYGIENE	LD	FB
**Time spent for lame management (hour/month)**	**[0; 7]**	**CONCRETE**	**TRIM**	**SCRAP>8**	**2**	**0**	**CONCRETE**	**TRIM**	**SCRAP>8**	**2**	**2**	**CONCRETE**	**TRIM**	**SCRAP>8**	**2**	**5**	**CONCRETE**	**TRIM**	**SCRAP>8**	**1**	**7**	**CONCRETE**	**TRIM**	**SCRAP≤8**	**1**	**7**
		**CONCRETE**	**TRIM**	**SCRAP>8**	**2**	**1**	**CONCRETE**	**TRIM**	**SCRAP>8**	**2**	**4**	**CONCRETE**	**TRIM**	**SCRAP>8**	**2**	**7**	**CONCRETE**	**TRIM**	**SCRAP>8**	**1**	**1**	**CONCRETE**	**TRIM**	**SCRAP≤8**	**1**	**3**
	**[7; 14]**	**CONCRETE**	**TRIM**	**SCRAP>8**	**4**	**0**	**CONCRETE**	**TRIM**	**SCRAP>8**	**5**	**4**	**CONCRETE**	**TRIM**	**SCRAP≤8**	**4**	**7**	**TEXTURED**	**TRIM**	**SCRAP>8**	**3**	**9**					
		**CONCRETE**	**TRIM**	**SCRAP>8**	**5**	**1**	**CONCRETE**	**TRIM**	**SCRAP>8**	**4**	**5**	**CONCRETE**	**TRIM**	**SCRAP≤8**	**4**	**8**	**TEXTURED**	**TRIM**	**SCRAP≤8**	**4**	**9**					
	**[14; 21]**	**CONCRETE**	**TRIM**	**SCRAP>8**	**6**	**1**	**CONCRETE**	**TRIM**	**SCRAP>8**	**6**	**5**	**CONCRETE**	**TRIM**	**SCRAP≤8**	**6**	**7**	**TEXTURED**	**TRIM**	**SCRAP≤8**	**6**	**9**					
		**CONCRETE**	**TRIM**	**SCRAP>8**	**6**	**0**	**CONCRETE**	**TRIM**	**SCRAP>8**	**6**	**7**	**CONCRETE**	**TRIM**	**SCRAP≤8**	**7**	**9**	**TEXTURED**	**NOTRIM**	**SCRAP≤8**	**7**	**9**					
	**[21; 28]**	**CONCRETE**	**TRIM**	**SCRAP>8**	**9**	**0**	**CONCRETE**	**TRIM**	**SCRAP>8**	**10**	**5**	**CONCRETE**	**TRIM**	**SCRAP≤8**	**8**	**9**						**TEXTURED**	**TRIM**	**SCRAP>8**	**1**	**7**
		**CONCRETE**	**TRIM**	**SCRAP>8**	**7**	**0**	**CONCRETE**	**TRIM**	**1**	**8**	**5**	**1**	**1**	**2**	**8**	**8**										
	**[28; 35]**	**CONCRETE**	**CONCRETE**	**1**	**11**	**0**	**1**	**1**	**1**	**11**	**7**	**1**	**2**	**2**	**11**	**9**										
		**CONCRETE**	**CONCRETE**	**1**	**11**	**1**	**1**	**1**	**1**	**11**	**9**	**1**	**2**	**2**	**10**	**9**										

A color code was applied to make the interpretation easier. Less desirable scenario 
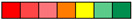
 More desirable scenario.

**Table 5 animals-13-01988-t005:** Optimal time to spend on footbath application according to the different combinations of scenarios.

Combination of Scenarios	Optimal Time (h/month)
CONCRETE_TRIM_SCRAP>8	22.2
CONCRETE_TRIM_SCRAP≤8	22.3
CONCRETE_NOTRIM_SCRAP>8	18.9
CONCRETE_NOTRIM_SCRAP≤8	19.0
TEXTURED_TRIM_SCRAP>8	18.8
TEXTURED_TRIM_SCRAP≤8	18.3
TEXTURED_NOTRIM_SCRAP>8	17.9
TEXTURED_NOTRIM_SCRAP≤8	17.8

CONCRETE, concrete floor; TEXTURED, textured floor; TRIM, preventive trimming; NOTRIM, no preventive trimming; SCRAP>8, daily scraping frequency >8 times; SCRAP≤8, daily scraping frequency ≤8 times.

**Table 6 animals-13-01988-t006:** Break-event point for hiring a farm employee according to the different combinations of scenarios.

Combination of Scenarios	Break-Even Point (h/month)
CONCRETE_TRIM_SCRAP>8	19.4
CONCRETE_TRIM_SCRAP≤8	19.8
CONCRETE_NOTRIM_SCRAP>8	17.4
CONCRETE_NOTRIM_SCRAP≤8	17.5
TEXTURED_TRIM_SCRAP>8	17.9
TEXTURED_TRIM_SCRAP≤8	17.1
TEXTURED_NOTRIM_SCRAP>8	16.1
TEXTURED_NOTRIM_SCRAP≤8	16.4

CONCRETE, concrete floor; TEXTURED, textured floor; TRIM, preventive trimming; NOTRIM, no preventive trimming; SCRAP>8, daily scraping frequency >8 times SCRAP≤8, daily scraping frequency ≤8 times.

## Data Availability

Data are available for consultation by contacting the author of correspondence.

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
