# Peer review of "Management of Digital Dermatitis in Dairy Herds: Optimization and Time Allocation"

_animals, 2023, doi:10.3390/ani13121988_

Round 1

Reviewer 1 Report

33: No it doesn't – “Time spent licking or grooming pen mates was similar for lame and nonlame cows; however, lame cows were more frequently the recipients of licking." Remove

54: Locomotion scoring detects lame cows earlier than farmers because it is systematic and standard farmer recognition is not. If you compare scoring by vets with scoring by farmers you don't get that difference. If farmers score they detect lameness at the same rate as trained scorers. It’s not about ability it's about what's done

62: the sensitivity is not the issue, you can test cows on an ongoing occasion twice a day, the issue is specificity

65: How can it be in countries where DD is rare or absent

74: having read the introduction I think you are confusing treatment for lameness with treatment for DD. They are not the same thing

208: do you mean <11 weeks

303: Your model assumes this so it's not surprising that this is what you find. Where is the sensitivity analysis that identifies what happens if this assumption is wrong

313: I don't think this is a relevant reference here

315: I think you need to justify your efficacy assumptions for footbathing - your original model used 5% copper sulphate based on one study when Thomson 2015 concluded after a systematic review determined that effectiveness of CuSO4 footbaths was not supported by evidence. Yes Solano

was after that review but it concluded that FB only worked when levels of DD were high

Reference 50 is the wrong reference 

54

Author Response

33: No it doesn't – “Time spent licking or grooming pen mates was similar for lame and nonlame cows; however, lame cows were more frequently the recipients of licking." Remove

AU: thank you for your comment. This detail was removed from the manuscript.

54: Locomotion scoring detects lame cows earlier than farmers because it is systematic and standard farmer recognition is not. If you compare scoring by vets with scoring by farmers you don't get that difference. If farmers score they detect lameness at the same rate as trained scorers. It’s not about ability it's about what's done

AU: The 5-scale lameness score classification developed by Sprecher et al., 1997 is a useful tool to evaluate the degree of lameness in cows. This classification is based on the observation of the posture and position of the 4 limbs. This method requires both time of observation but also experience and some technical knowledge to classify the cows well. The authors meant that depending on the skills of the farmers and the time allocated to lameness detection, the detection of different levels of lameness is not equal from one farm to another. In fact, one study led by Alawneh et al. (2012) showed that the standard detection rate of lame cows was highly dependent on the lameness score and was not optimal for intermediate scores. The previous study from Robcis et al. (2023) built on Alawneh's results and divided the detection rate into 11 detection levels to accurately simulate the different behaviors of farmers in detecting lame cows. This aspect was clarified in the manuscript, please see lines 57-63

62: the sensitivity is not the issue, you can test cows on an ongoing occasion twice a day, the issue is specificity

AU: thank you for your comment. Authors agree with you and bring an appropriate correction in the manuscript

65: How can it be in countries where DD is rare or absent

AU: The management of lameness in dairy herds includes both the management of infectious foot diseases (DD, ID and IP to a lesser extent) and non-infectious foot diseases (SU and WLD). Some countries or regions are fortunate to have very few or no cases of DD. In the where DD cases are rare, individual treatment is recommended by trimming claw and applying oxytetracycline spray, possibly with a bandage on top. In case of DD-free regions, the management of lameness is only focused on other foot disease management. This still includes detection of lame cows and regular trimming sessions. Risk factor management is also a key point, as shown by Robcis et al. (2023).

74: having read the introduction I think you are confusing treatment for lameness with treatment for DD. They are not the same thing

AU: The present study deals with lameness management, with the consideration of 5 etiologies of lameness. The different treatment strategies were implemented in the preliminary study conducted by Robcis et al. (2023) for each etiology. The treatment strategy for DD is a special case, since it includes both individual and collective treatments, the decision depending on the level of DD prevalence within the herd. In the previous study led by Robcis et al. (2023), DD accounted for the largest share of lameness, as has been the case in other field studies. Moreover, Robcis et al. (2023) showed that total lameness cases and MLD was highly correlated to DD prevalence. Eradication of this disease is impossible and management strategy is a key issue in dairy herds. For this reason, specific attention has been given to the specific management of DD. However, other foot diseases and their therapeutic management are also included in the present study, according to a holistic and integrative approach

208: do you mean <11 weeks

AU: Yes. The manuscript was consequently modified to bring more clarity

303: Your model assumes this so it's not surprising that this is what you find. Where is the sensitivity analysis that identifies what happens if this assumption is wrong

AU: The authors assume that they performed the present study without preconceived conclusions. As described in the previous study led by Robcis et al. (2023), the model used was built according to a stochastic approach and a simulated stabilized herd was used as a unique starting base for all scenarios for a 728-wk simulation with 100 iterations each. The last 520 wk were included in the results analysis to obtain stable results. Some sentences were added about that subject in the manuscript, please see lines 117-122

313: I don't think this is a relevant reference here

AU: thank you for your remark. The right reference was added in the manuscript

315: I think you need to justify your efficacy assumptions for footbathing - your original model used 5% copper sulphate based on one study when Thomson 2015 concluded after a systematic review determined that effectiveness of CuSO4 footbaths was not supported by evidence. Yes Solano

was after that review but it concluded that FB only worked when levels of DD were high

AU: The therapeutic and preventive efficacy of footbaths in digital dermatitis cases are widely discussed in the literature. The calibration of our initial model was based on the meta-analysis led by Jacobs et al. (2019), showing that the use of a footbath with CuS04 used at least 4 times per week was the only effective strategy in the curative treatment of DD lesions. Preventive efficacy was not shown, that is why it was not included in our model. Some sentences were added in the manuscript to discuss this point, please see lines 354-357

Reference 50 is the wrong reference 

AU: The right reference was added

Reviewer 2 Report

Management of Lameness in Dairy Herds: Optimization and Time Allocation

This study is an interesting approach to the early detection of lameness, a common problem inside dairy farms. Due to the economic implications that this pathology causes, it is important to keep studying methods to prevent the affectation on the welfare of animals. I left some comments below hoping they can be helpful for the authors.

Lines 9-13: I suggest adding the aim of the study in the simple summary and improving the conclusion.

Lines 14 – 15: Please, include in the aim of the study that it was performed in dairy herds. In line 75, when stating the objective in the introduction, please, do the same.

Line 28: The authors could include “animal welfare” as a keyword.

 Line 33: If the authors mean “severe pain” with “higher pain”, I recommend using the first term.

 Line 45: Please, define the abbreviation of “DD”.

Line 56: Delete the parenthesis before the references.

Line 62: It would be adequate to include some examples of the image-processing techniques that are currently applied to detect lameness. This articles might help: https://doi.org/10.3390/s150614513

 https://doi.org/10.1079/cabireviews202217040

 Line 76: If the authors had a hypothesis for the study, I recommend adding it.

 Line 80: I recommend adding more information regarding the simulation model. For example, the age of the animals, year season,  sex, and reproductive stage.

 Line 85. Amend in-text citation.

 Lines 303-305: If possible, add a percentage depicting the incidence of lameness due to digital dermatitis in dairy cows. Also, additional information regarding if it is the first clinical sign of the disease or if is related to infectious agents would enrich the discussion.

Line 309-313: Here, it would be interesting to also mention the physiological consequences that arise when lameness is not addressed promptly. For example, the inflammatory response, acute pain that can transform into chronic pain when proper treatment is not provided are elements that need to be addressed.

 Line 323: Please, insert a reference.

 Line 375: Briefly mention some limitantions of the study.

 Reference list: Revise the template of the journal and amend the reference list (e.g., Author 1, A.B.; Author 2, C.D. Title of the article. Abbreviated Journal Name Year, Volume, page range.)

Author Response

This study is an interesting approach to the early detection of lameness, a common problem inside dairy farms. Due to the economic implications that this pathology causes, it is important to keep studying methods to prevent the affectation on the welfare of animals. I left some comments below hoping they can be helpful for the authors.

Lines 9-13: I suggest adding the aim of the study in the simple summary and improving the conclusion.

AU: thank you for your comment. The simple summary was modified according to your suggestions, please see lines 11-16

Lines 14 – 15: Please, include in the aim of the study that it was performed in dairy herds. In line 75, when stating the objective in the introduction, please, do the same.

AU: your suggestions were added at the appropriate place in the manuscript

Line 28: The authors could include “animal welfare” as a keyword.

AU: the suggested keyword was added, please see line 31

 Line 33: If the authors mean “severe pain” with “higher pain”, I recommend using the first term.

AU: thank you for your comment. Your suggestion was replaced in the manuscript, please see line 36

 Line 45: Please, define the abbreviation of “DD”.

AU: Done.

Line 56: Delete the parenthesis before the references.

AU: Done.

Line 62: It would be adequate to include some examples of the image-processing techniques that are currently applied to detect lameness. This articles might help: 

https://doi.org/10.3390/s150614513

 https://doi.org/10.1079/cabireviews202217040

AU: thank you for your comment. This technique was added in the manuscript with the supportive references you suggested, please see lines 71-74

Line 76: If the authors had a hypothesis for the study, I recommend adding it.

AU: At the beginning of the study, the authors had no preferred hypothesis and analyzed the data provided by the model without preconceived ideas

Line 80: I recommend adding more information regarding the simulation model. For example, the age of the animals, year season, sex, and reproductive stage.

AU: thank you for your comment. Details were added in the manuscript, please see lines 90-122

Line 85. Amend in-text citation.

AU: the citation was corrected

Lines 303-305: If possible, add a percentage depicting the incidence of lameness due to digital dermatitis in dairy cows. Also, additional information regarding if it is the first clinical sign of the disease or if is related to infectious agents would enrich the discussion.

AU: thank you for your comment. Some sentences were added in the manuscript, please see lines 321-328

Line 309-313: Here, it would be interesting to also mention the physiological consequences that arise when lameness is not addressed promptly. For example, the inflammatory response, acute pain that can transform into chronic pain when proper treatment is not provided are elements that need to be addressed.

AU: thank you for your very interesting comment. Consideration of chronic lameness was not included in this model and this is a limitation of the present study. This aspect was consequently mentioned in the manuscript and some sentences were added about this subject, please see lines 337-347

Line 323: Please, insert a reference.

AU: Done.

 Line 375: Briefly mention some limitations of the study.

AU: A limitation of the study was already in the manuscript (spillover effect, please see lines 363-370). Some other limitations were added at the end of the discussion, please see lines 415-419

Reference list: Revise the template of the journal and amend the reference list (e.g., Author 1, A.B.; Author 2, C.D. Title of the article. Abbreviated Journal Name YearVolume, page range.)

AU: The references were indicated with the right layout

Round 2

Reviewer 2 Report

The authors have substantially improved the manuscript. They have followed all my advice and responded step by step.

I have no additional comments.

The article must be published

Author Response

Authors warmly thank you for your suggestions improving the manuscript.